# Diverse Message Passing for Attribute with Heterophily

**Liang Yang** [1,2,*]**, Mengzhe Li** [1,*]**, Liyang Liu** [1,*]
**Bingxin Niu** [1]**, Chuan Wang** [2]**, Xiaochun Cao** [3]**, Yuanfang Guo** [4,†]
[1]School of Artificial Intelligence, Hebei University of Technology, Tianjin, China
[2]State Key Laboratory of Information Security, IIE, CAS, Beijing, China
[3]School of Cyber Science and Technology, Sun Yat-sen University, Shenzhen, China
[4]State Key Laboratory of Software Development Environment, Beihang University, China
yangliang@vip.qq.com, limengzhefree@foxmail.com, liyang_liu1@163.com
niubingxin666@163.com, {wangchuan,caoxiaochun}@iie.ac.cn, andyguo@buaa.edu.cn

## Abstract

Most of the existing GNNs can be modeled via the Uniform Message Passing framework. This framework considers all the attributes of each node in its entirety, shares the uniform propagation weights along each edge, and focuses on the uniform weight learning. The design of this framework possesses two prerequisites, the simplification of homophily and heterophily to the node-level property and the ignorance of attribute differences. Unfortunately, different attributes possess diverse characteristics. In this paper, the network homophily rate defined with respect to the node labels is extended to attribute homophily rate by taking the attributes as weak labels. Based on this attribute homophily rate, we propose a Diverse Message Passing (DMP) framework, which specifies every attribute propagation weight on each edge. Besides, we propose two specific strategies to significantly reduce the computational complexity of DMP to prevent the overfitting issue. By investigating the spectral characteristics, existing spectral GNNs are actually equivalent to a degenerated version of DMP. From the perspective of numerical optimization, we provide a theoretical analysis to demonstrate DMP's powerful representation ability and the ability of alleviating the over-smoothing issue. Evaluations on various real networks demonstrate the superiority of our DMP on handling the networks with heterophily and alleviating the over-smoothing issue, compared to the existing state-of-the-arts.

## 1 Introduction

Originated from the spectral graph theory, Graph Neural Networks (GNNs)[1, 2] have demonstrated superior performances on modeling ubiquitous irregular data in many tasks, such as computer vision, natural language processing, information retrieval, etc. By reducing the computational complexity of graph Fourier transformation via approximation, most of the GNNs, especially Graph Convolutional Networks (GCN) [3] and its variants (such as SGC [4]), actually perform Laplacian smoothing [5] and low-passing filtering [4]. Although this intuitive strategy achieves state-of-the-art performances on networks with homophily, such as Cora, CiteSeer and Pubmed, it possesses two fatal drawbacks. Firstly, it only performs well on shallow networks, while induces serious over-smoothing issue when more layers are stacked [5, 6, 7]. Secondly, it shows obvious ineffectiveness in handling the networks

---

[*]Equal contribution.
[†]Corresponding author.

35th Conference on Neural Information Processing Systems (NeurIPS 2021).

with heterophily, where its performance is even worse than the simple multiple layer perceptron (MLP) [8].

There are many attempts to overcome the above two drawbacks. The common philosophy behind these attempts is weakening the smoothing effect. One type of methods directly modifies the smoothed node representations based on local information. For example, APPNP [9] and GCNII [10] average the smoothed node representations with the corresponding original node attributes. JKNet [7] and PPNP [9] combine multi-scale information by summation, while MixHop [11], truncated block Krylov network [12] and H2GCN [8] combine multi-scale information by concatenation. Other methods modifies the network topology, which indirectly reduces the over-smoothing issue. For instance, DropEdge [13] and GRAND [14] randomly remove edges and nodes, respectively, while FAGCN [15] and GPRGNN[16] relax the edges with positive values, which tends to induce the smoothing effect, to possess real values (which can be either positive or negative). A recent research [17] has tried to bridge and unify over-smoothing issue and networks with heterophily.

Most of the GNNs, which can be designed for networks with either homophily or heterophily, can be unified by a message passing framework. These GNNs usually consider all the attributes of one node in its entirety, and propagate them via the same (positive or negative) propagation weight along each edge. Intuitively, most of the Graph Neural Networks (GNNs) are restricted in the *uniform* message passing framework and focus on uniform weight learning, such as GCN [3], GAT [18] and FAGCN [15]. This phenomenon is induced by simplifying the network homophily/heterophily to node-level property [19]. In Geom-GCN [20] and H2GCN [8], which are designed to handle the networks with heterophily, the network homophily rate is defined by only utilizing network topology and node labels.

Actually, different attributes possess diverse characteristics. To explicitly model the diversity, the concept of network homophily rate defined in [20] is extended to attribute level. Specifically, attribute homophily rate for each attribute is defined by replacing the real label of each node with the corresponding attributes to form weak labels. Statistics in Figure 1 reveal that the attribute homophily rates of the networks with either homophily and heterophily all tend to vary in a large range. Then, GNNs are desired to explicitly consider the diversity of attribute homophily rates. To achieve this objective, we propose a Diverse Message Passing (DMP) framework, which specifies every attribute propagation weight on each edge, to adaptively handle the diverse attribute homophily. Besides, we further propose two strategies to significantly reduce the model complexity of DMP to alleviate the overfitting issue. By investigating the spectral characteristics, existing spectral GNNs are actually equivalent to a degenerated version of our DMP. From the perspective of numerical optimization, we provide a theoretical analysis to demonstrate DMP's powerful representation ability and the ability of alleviating the over-smoothing issue. Specifically, different from Uniform Message Passing, which directly generates a graph partition, our DMP generates multiple groups of graph partition candidates, and then the classifier in semi-supervised task will determine the specific combining approach to form the final partition. This mechanism actually enables our DMP to possess more representation ability, which can also alleviate the over-smoothing issue, compared to classic Uniform Message Passing.

## 2 Notations and Preliminaries

Let $\mathcal{G} = (\mathcal{V}, \mathcal{E})$ denote a graph with a set of nodes $\mathcal{V} = \{v_1, v_2, \cdots, v_N\}$ and a set of edges $\mathcal{E}$, where $N$ is the number of nodes. The topology of graph $\mathcal{G}$ can be represented by its adjacency matrix $\mathbf{A} = [a_{ij}] \in \{0, 1\}^{N \times N}$, where $a_{ij} = 1$ if and only if there exists an edge $e_{ij} = (v_i, v_j)$ between nodes $v_i$ and $v_j$. The degree matrix $\mathbf{D}$ is a diagonal matrix with each diagonal element $d_i = \sum_{i=1}^{N} a_{ij}$ being the degree of node $v_i$. $\mathcal{N}(v_i) = \{v_j | (v_i, v_j) \in \mathcal{E}\}$ stands for the neighbourhoods of node $v_i$. $\mathbf{X} \in \mathbf{R}^{N \times F}$ and $\mathbf{H} \in \mathbf{R}^{N \times F'}$ respectively denote the collection of node attributes and representation of the $i^{th}$ row, i.e., $\mathbf{x}_i \in \mathbb{R}^F$ and $\mathbf{h}_i \in \mathbb{R}^{F'}$, which both correspond to node $v_i$. Note that $F$ and $F'$ stand for the dimensions of node attributes and representations.

Most of the Graph Neural Networks (GNNs) follow an aggregation-combination strategy [21], where each node representation is iteratively updated by aggregating node representations in the local neighbourhoods and combining the aggregated representations with the node representation itself as

$$\bar{\mathbf{h}}_v^k = \text{AGGREGATE}^k \left( \{ \mathbf{h}_u^{k-1} | u \in \mathcal{N}(v) \} \right), \quad \mathbf{h}_v^k = \text{COMBINE}^k \left( \mathbf{h}_v^{k-1}, \bar{\mathbf{h}}_v^k \right), \quad (1)$$

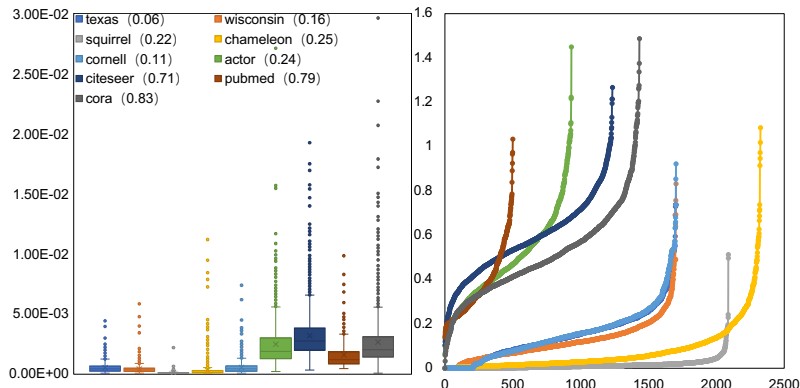

Figure 1: Attribute homophily rates on networks with different homophily. Each number, which corresponds to a network name, denotes the network homophily rate computed via the method in [20]. (a) Distributions of attribute homophily rates, i.e., $\{\beta_f\}_{f=1}^F$, on different networks. Each column represents one network. (b) Attribute homophily rate, i.e., $\{\beta_f\}_{f=1}^F$, on different networks. Each curve denotes one network. For better visualization, attribute homophily rates are sorted in ascending order.

where $\bar{\mathbf{h}}_v^k$ stands for the aggregated representation from local neighbourhoods. Besides of the concatenation based implementation, such as GraphSAGE [22] and H2GCN [8], averaging (or summation) has been widely adopted to implement $\text{COMBINATE}^k(\cdot, \cdot)$, such as GCN [3], GAT [18], GIN [23], etc. Except for the MAX and LSTM implementations in GraphSAGE [22], most of the GNNs utilize averaging function to implement $\text{AGGREGATE}^k$. Therefore, they can be unified as

$$\mathbf{h}_v^k = \sigma\left(\left(c_{vv}^k \mathbf{h}_v^{k-1} + \sum_{u \in \mathcal{N}(v)} c_{uv}^k \mathbf{h}_u^{k-1}\right) \mathbf{W}^k\right), \tag{2}$$

where $\mathbf{W}^k$ represents the learnable parameters and $\sigma(\cdot)$ denotes the nonlinear mapping function. Note that the scalar $c_{uv}$ is the averaging weight. For example, GCN [3] sets $c_{uv}^k = 1/(\sqrt{(d_u + 1)(d_v + 1)})$, GIN [23] sets $c_{uv}^k = 1$ for $u \neq v$ and $c_{vv}^k = 1 + \epsilon^k$, and GAT [18] learns non-negative $c_{uv}^k$ based on the attention mechanism. Recently, to handle the network with heterophily via high-passing filtering, GPRGNN [16] sets $c_{uv}^k = \gamma^k/(\sqrt{(d_u + 1)(d_v + 1)})$ with $\gamma^k$ being a learnable real value, while FAGCN [15] directly relaxes the learnable $c_{uv}$ in GAT to real value.

## 3 Diverse Message Passing

In this section, the motivation of Diverse Message Passing is firstly provided. Then, the diverse aggregation framework is given, followed by two implementations to reduce model complexity.

### 3.1 Motivations

The unified formulation in Eq. (2) can be regarded as the *uniform* message passing framework, where all the attributes which are propagated from node $v$ to node $u$, i.e., all the elements of $\mathbf{h}_u^{k-1}$, share the common weight $c_{uv}^k$. Then, all the attributes can be aggregated with the same weight assignment scheme in the local neighbourhood $\mathcal{N}(v) \cup \{v\}$. Since aggregation can be interpreted from the perspectives of both Laplacian smoothing [5] and low-passing filtering [4], the same weight assignment scheme indicates that all the attributes tend to share similar distributions over the entire graph.

To investigate the distributions of attributes and measure their similarities, the attribute homophily rate is proposed. The network homophily rates, including both the node version [20] and edge version [8], quantify similarity between connected nodes. The node similarity in network homophily rate is measured according to the node labels. Here, the attribute homophily rate is defined by measuring

the node similarity according to each attribute. Specifically, the attribute homophily rate of attribute $f$ is formulated as

$$\beta_f = \frac{1}{\sum_{v \in \mathcal{V}} x_{vf}} \sum_{v \in \mathcal{V}} \beta_{vf} = \frac{1}{\sum_{v \in \mathcal{V}} x_{vf}} \sum_{v \in \mathcal{V}} \left( x_{vf} \frac{\sum_{u \in \mathcal{N}(v)} x_{uf}}{d_v} \right), \tag{3}$$

where $\frac{\sum_{u \in \mathcal{N}(v)} x_{uf}}{d_v}$ is the rate of neighbourhoods with attribute $f$, $\beta_{vf} = x_{vf} \frac{\sum_{u \in \mathcal{N}(v)} x_{uf}}{d_v}$ is the rate of neighbourhoods which share the same attribute $f$ as node $v$, and the attribute homophily rate of attribute $f$, i.e., $\beta_f$, is the average of $\beta_{vf}$ over all the nodes possessing attribute $f$.

Figure 1 shows the distribution of $\{\beta_f\}_{f=1}^F$ on networks with different homophily, computed via the method in [20]. For better visualization, both the macroscopic and microscopic perspectives are provided. Figure 1(a) gives the distribution of attribute homophily (in macroscopic perspective) on each network (each column). Figure 1(b) describes the attribute homophily of different attributes, i.e., $\{\beta_f\}_{f=1}^F$ from the microscopic perspective.

It can be observed that networks with higher homophily also possess higher averaged attribute homophily, and vice versa. However, attribute homophily $\{\beta_f\}_{f=1}^F$ distributes in a large range, with little correlations to the degree of attribute homophily of network. It shows that different attributes possess remarkably different attribute homophily, which reveals that the distributions of attributes over the entire graph are diverse, instead of uniform ones. Therefore, diverse aggregation scheme, i.e., averaging scheme, should be designed to fully explore the diverse distributions of attributes over the entire graph for effective node representation learning.

## 3.2 Diverse Aggregation Framework

To achieve diverse aggregation, the Uniform Message Passing in Eq. (2) is revised to make different attributes possess different propagation weights. Then, the scaler $c_{vu}^k$ in Eq. (2) is augmented to vector $\mathbf{c}_{vu}^k$, which has the same length as the node representation $\mathbf{h}_u^{k-1}$. Each element in $c_{vu}^k$ stands for the weight of the corresponding attribute in $\mathbf{h}_u^{k-1}$ propagated from node $u$ to node $v$. Thus, Eq. (2) is modified to

$$\mathbf{h}_v^k = \sigma \left( \left( \mathbf{c}_{vv}^k \odot \mathbf{h}_v^{k-1} + \sum_{u \in \mathcal{N}(v)} \mathbf{c}_{uv}^k \odot \mathbf{h}_u^{k-1} \right) \mathbf{W}^k \right), \tag{4}$$

where $\odot$ denotes the element-wise product of vectors. Then, the remaining issue is the learning strategy of $\mathbf{c}_{uv}^k$'s. If all $\mathbf{c}_{uv}^k$'s are set as free parameters to be directly learned, the model complexity, i.e., the number of parameters, of directly learning $\mathbf{c}_{uv}^k$'s is $\mathcal{O}(|\mathcal{E}|F)$, which may induce overfitting. In the following paragraphs, two strategies are designed to reduce the number of learnable parameters.

**The first strategy** constrains all the parameters $\mathbf{c}_{uv}^k$'s to be determined by a learnable function. Since $\mathbf{c}_{uv}^k$ can be regarded as the averaging weight between nodes $v$ and $u$, the attention scheme can be adopted as

$$\mathbf{c}_{uv}^k = tanh \left( [\mathbf{h}_v^{k-1} || \mathbf{h}_u^{k-1}] \mathbf{W}_c^k \right), \tag{5}$$

where $[\mathbf{h}_v^{k-1} || \mathbf{h}_u^{k-1}]$ represents the concatenation of representations $\mathbf{h}_v^{k-1}$ and $\mathbf{h}_u^{k-1}$. Note that the learnable mapping function $\mathbf{W}_c^k \in \mathbb{R}^{F \times F}$ is shared across all the edges, where $F$ is the dimension of $\mathbf{h}_v^{k-1}$ and $\mathbf{c}_{uv}^k$. Then, the model complexity is significantly reduced to $\mathcal{O}(F^2)$. Different from the attention mechanism employed in GAT [18], which constrains the learned propagation weights $c_{uv}^k$'s to be non-negative via a softmax nonlinear function, the learned weight vectors $\mathbf{c}_{uv}^k$'s are relaxed to real values by adopting a tanh nonlinear function $tanh(x) = \frac{\exp(x) - \exp(-x)}{\exp(x) + \exp(-x)} = sigmoid(2x) - 1$, whose output is zero-centered and ranged in $(-1, 1)$. Intuitively, the positive weights are equivalent to low-passing filtering, while the negative weights actually facilitate the filtering beyond low-frequency.

**The second strategy** simplifies Eq. (4) by assuming that all the neighbourhoods employs identical propagation vector, i.e., $\mathbf{c}_{uv}^k = \bar{\mathbf{c}}_v^k$ for every $u \in \mathcal{N}(v)$. This simplification, which has also been adopted in GraphSAGE [22], H2GCN [8] and GIN[23], assumes that the neighbourhoods of fixed-hop equally contribute to the node representations, Accordingly, Eq. (4) can be simplified to

$$\mathbf{h}_v^k = \sigma \left( \left( \mathbf{c}_v^k \odot \mathbf{h}_v^{k-1} + \bar{\mathbf{c}}_v^k \odot \sum_{u \in \mathcal{N}(v)} \mathbf{h}_u^{k-1} \right) \mathbf{W}^k \right). \tag{6}$$

Eq.(6) can be considered as the weighted averaging of the node representation $\mathbf{h}_v^{k-1}$ and the neighbourhood representation $\bar{\mathbf{h}}_v^{k-1} = \sum_{u \in \mathcal{N}(v)} \mathbf{h}_u^{k-1}$, where both $\mathbf{c}_v^k$ and $\bar{\mathbf{c}}_v^k$ can be regarded as the element-wise averaging weights. Thus, the learning of $\mathbf{c}_v^k$ and $\bar{\mathbf{c}}_v^k$ can be determined by node and neighbourhood representations. Similar to Eq. (5), the attention mechanism is adopted to learn them as

$$\mathbf{c}_v^k = tanh\left([\mathbf{h}_v^{k-1}||\bar{\mathbf{h}}_v^{k-1}]\mathbf{W}_c^k\right). \tag{7}$$

Compared to Eq.(5), the learning process of $\bar{\mathbf{c}}_v^k$ is similar, yet with a different parameter $\mathbf{W}_{\bar{c}}^k$. This strategy possesses identical model complexity to the first strategy.

## 3.3 Interpretation from the Spectral Perspective

In previous subsections, Diverse Message Passing is motivated and proposed from the spatial perspective of GNNs. Actually, GNNs originate from spectral graph theory, where ChebNet [24] and GCN [3] are two representative methods. ChebNet utilizes the Chebyshev polynomials of Laplacian matrix to define the graph convolution as $\mathbf{H} = \sum_{r=0}^{R} T_r(\tilde{\mathbf{L}})\mathbf{X}\mathbf{W}_r$, where $\tilde{\mathbf{L}}$ denotes the scaled Laplacian matrix and $T_r(x)$ represents the Chebyshev polynomials. GCN simplifies ChebNet by constraining it to a 1-order model, i.e., $R = 1$, as

$$\mathbf{h}_v^k = \sigma\left(\mathbf{h}_v^{k-1}\mathbf{W}_0^k + \left(\sum_{u \in \mathcal{N}(v)} \mathbf{h}_u^{k-1}\right)\mathbf{W}_1^k\right). \tag{8}$$

Then,GCN force the two terms in Eq.(8) to share the same mapping function, i.e., $\mathbf{W}_0^k = \mathbf{W}_1^k = \mathbf{W}^k$. It can be observed that if all the nodes share the same $\mathbf{c}_v^k$ and $\bar{\mathbf{c}}_v^k$ as $\mathbf{c}^k$ and $\bar{\mathbf{c}}^k$, the second strategy in Eq. (6) degrades to a 1-order ChebNet in Eq. (8) with $\mathbf{W}_0^k = (\mathbf{1}\cdot\mathbf{c}^k)\odot\mathbf{W}^k$ and $\mathbf{W}_1^k = (\mathbf{1}\cdot\bar{\mathbf{c}}^k)\odot\mathbf{W}^k$, where $\mathbf{1}$ stands for the vector of ones and it possesses the same length as $\mathbf{c}^k$ and $\bar{\mathbf{c}}^k$. Therefore, the mapping function $\mathbf{W}$ in both ChebNet and GCN contain the attribute-wise propagation weights, $\mathbf{c}^k$ and $\bar{\mathbf{c}}^k$, which are shared by all the edges. Therefore, most of the spectral GNNs tend to fail in the Uniform Message Passing framework, which has also been explained in [25].

In fact, the attribute-wise weights $\mathbf{c}_{uv}^k$ in Eq. (4) and $\mathbf{c}_v^k$ in Eq. (6) cannot be integrated into the mapping parameter $\mathbf{W}^k$, because these weights tend to be different, on different edges in Eq. (4) and different nodes in Eq. (6), to model the attribute heterophily. Hence, Diverse Message Passing may break the performance ceiling of spectral GNNs.

## 3.4 Theoretical Analysis

Recently, some efforts have been paid to interpreted GNNs from the perspective of numerical optimization [26, 27, 28]. Each graph convolutional layer in GCN [3] can be regarded as a gradient descent algorithm to minimize the graph Laplacian regularization $tr(\mathbf{H}^T\tilde{\mathbf{L}}\mathbf{H}) = \frac{1}{2}\sum_{u,v}\|\mathbf{h}_u - \mathbf{h}_v\|_2^2$, where $tr(\cdot)$ stands for the trace of the matrix, starting from the node attribute $\mathbf{X}$. However, most of the existing literatures only unify and interpret GNNs based on the fixed propagation weight in Eq. (4), such as APPNP [9], JKNet [7] and DAGNN [29].

In this subsection, the theoretical difference between the Uniform Message Passing in Eq. (2) and Diverse Message Passing in Eq. (4) is analyzed. Since the propagation weights on all the attributes should be learnable to facilitate the Diverse Message Passing, the previous numerical optimization framework for GNNs with fixed propagations is desired to be relaxed. Specifically, the propagation weights $c_{uv}^k$ in Eq. (2) and $\mathbf{c}_{uv}^k$ in Eq. (4) are both assumed to be learnable. Since this analysis focuses on the underlying philosophy of learning propagation weights from the edge-wise/channel-wise perspective, the specific learning function is neglected.

**Uniform Message Passing** in Eq. (2) with learnable weights $c_{uv}^k$ can be regarded as the combination of two steps: 1) learning the propagation weights to form a new graph topology; 2) propagating attributes on the new graph. Then, the following theorem holds.

**Theorem 1.** *The Uniform Message Passing in Eq. (2) with learnable weights $c_{uv}$ is the gradient descent algorithm of the following objective function with node attribute $\mathbf{X}$ being the initialization of*

**H**.

$$\min_{\mathbf{C},\mathbf{H}} \sum_{u,v} \left(b_{uv}c_{uv} + \gamma c_{uv}^2\right) + 2tr(\mathbf{H}^T \mathbf{L}_C \mathbf{H}), \tag{9}$$

$$s.t. \ \forall u \sum c_{uv} = 1, \ 0 \le c_{uv} \le 1, \ \mathbf{H} \in \mathbf{R}^{N \times F}, \tag{10}$$

*where $b_{uv} = g\left(a_{uv}, dis(\mathbf{x}_i, \mathbf{x}_j)\right)$ denotes the similarity between nodes $u$ and $v$, according to both the topology $a_{uv}$ and the distance between attributes $dis(\mathbf{x}_i, \mathbf{x}_j)$. $\mathbf{A} = [a_{uv}]$ is the adjacency matrix of $\mathcal{G}$. $\mathbf{C}$ represents the collection of $c_{uv}$, i.e., the adjacency matrix of the learned graph. $\mathbf{L}_C$ stands for the Laplacian matrix of the adjacency matrix $\mathbf{C}$.*

Theorem 1 can be easily proved. The sketch of the proof is that the minimization of Eq. (9) can be performed by alternately updating $c_{uv}$ and $\mathbf{H}$, which correspond to the two step of Uniform Message Passing. The detailed proof is provided in Appendix.

To further understand the insight of the objective function of Uniform Message Passing, two important theorems are provided. They bridge the graph Laplacian regularization, eigenvalue of the Laplacian matrix and the number of connected components in the graph.

**Theorem 2.** *[Ky Fan's Theorem [30]] There exists*

$$\min_{\mathbf{H} \in \mathbf{R}^{N \times F}, \mathbf{H}^T \mathbf{H} = \mathbf{I}} tr(\mathbf{H}^T \mathbf{L}_C \mathbf{H}) = \sum_{f=1}^{F} \sigma_f(\mathbf{L}_C), \tag{11}$$

*where $\sigma_f(\mathbf{L}_C)$ denotes the $f^{th}$ smallest eigenvalue of the Laplacian matrix $\mathbf{L}_C$.*

**Theorem 3.** *[[31, 32]] The multiplicity $F$ of the eigenvalue 0 of the Laplacian matrix $\mathbf{L}_C$ equals to the number of connected components in the graph, whose similarity matrix is $\mathbf{C}$.*

Theorem 3 reveals that the graph can be partitioned into $\mathbf{F}$ connected components, if $rank(\mathbf{L}_C) = N - F$. Then, as shown in the following theorem, Uniform Message Passing actually performs graph partitions based on the similarity between nodes, which is calculated with respect to both the topology and attribute distances.

**Theorem 4.** *The Uniform Message Passing in Eq. (2) actually partitions graph into $F$ connected components based on the similarity $b_{uv} = g\left(a_{uv}, dis(\mathbf{x}_i, \mathbf{x}_j)\right)$ via*

$$\min_{\mathbf{C}} \sum_{u,v} \left(b_{uv}c_{uv} + \gamma c_{uv}^2\right) \tag{12}$$

$$s.t. \ \forall u \sum c_{uv} = 1, \ 0 \le c_{uv} \le 1, \ rank(\mathbf{L}_C) = N - F. \tag{13}$$

**Diverse Message Passing** in Eq. (4) with learnable weights $\mathbf{c}_{uv}^k$ extends Uniform Message Passing by propagating different attributes with different weights. By following the derivation of Theorem 4, we can obtain the following theorem for Diverse Message Passing.

**Theorem 5.** *The Diverse Message Passing in Eq. (4) actually partitions graph into $2$ connected components ($F$ groups) based on each similarity $b_{uv}^{(f)} = g\left(a_{uv}, dis(x_{if}, x_{jf})\right)$ via*

$$\min_{\mathbf{C}^{(f)}} \sum_{u,v} \left(b_{uv}^{(f)}c_{uv}^{(f)} + \gamma (c_{uv}^{(f)})^2\right), f = 1, ..., F. \tag{14}$$

$$s.t. \ \forall u \sum c_{uv}^{(f)} = 1, \ 0 \le c_{uv}^{(f)} \le 1, \ rank(\mathbf{L}_C^{(f)}) = N - 2. \tag{15}$$

Different from Uniform Message Passing, which directly generates a $F$-components partition, Diverse Message Passing generates $F$ groups of 2-components partitions as candidates. Then the classifier in semi-supervised task determines how to combine them to form the final $F$-components partition. This shows that Diverse Message Passing possesses a superior representation ability, compared to classic Uniform Message Passing.

### 3.4.1 Alleviating the Oversmoothing Issue

The powerful representation ability of Diverse Message Passing also indicates that DMP can alleviate the over-smoothing issue. According to Theorem 1 in [5], the over-smoothing issue will force all the

node embeddings to converge to a few vectors, which are fully determined by the indication vectors of the connected components and thus make the nodes indistinguishable. In the following example, we will demonstrate that the number of converged vectors from our Diverse Message Passing is much larger than that from Uniform Message Passing, which reveals that our DMP can effectively alleviate the over-smoothing issue.

Given a graph $\mathcal{G}$ with $k$ connected components $\{C_i\}_{i=1}^k$, the indication vector for the $i$-th component is denoted by $\mathbf{1}^{(i)} \in \{0,1\}^k$. This vector indicates the relationship between a vertex and the component $C_i$, i.e.,

$$\mathbf{1}_j^{(i)} = \left\{ \begin{array}{l} 1, v_j \in C_i \\ 0, v_j \notin C_i. \end{array} \right. \tag{16}$$

According to Theorem 4, the Uniform Message Passing actually partitions the graph $\mathcal{G}$ into $F$ connected components. Then, the embeddings in Uniform Message Passing will converge to a one-hot vector $\mathbf{1}^{(i)} \in \{0,1\}^k$, where only one element is 1 and the other elements are 0, for $i = 1, \ldots, F$, i.e., the over-smoothing issue appears.

On the contrary, Theorem 5 reveals that our Diverse Message Passing generates $F$ groups of 2-components partitions, which is equivalent to generating $2^F$ candidate components. If the 2-components partition of the j-th group for node $i$ is represented by a scalar $t_j^{(i)} = \{0,1\}$, the $F$ groups of 2-components partitions for node $i$ can be represented by $\mathbf{t}^{(i)} = \{0,1\}^k$. Note that different from $\mathbf{1}^{(i)} \in \{0,1\}^k$ in Uniform Message Passing, where only one element is 1 and the other elements are 0, each element in $\mathbf{t}^{(i)} = \{0,1\}^k$ can be either 1 or 0. Then, $\mathbf{t}^{(i)} = \{0,1\}^k$, to which the embeddings in our Diverse Message Passing tend to converge, may possess $2^F$ different values.

The number of different values in $\mathbf{t}^{(i)} = \{0,1\}^k$ is $2^F$, which is apparently much larger than that in $\mathbf{1}^{(i)} \in \{0,1\}^k$, i.e., $F$. Therefore, the number of converged values from our Diverse Message Passing is much larger than that from Uniform Message Passing, which proves that our Diverse Message Passing can effectively alleviate the over-smoothing issue.

For intuitive illustration, a toy example with $F = 3$ is provided here. The embeddings generated from Uniform Message Passing will converge to $(1,0,0)$, $(0,1,0)$ and $(0,0,1)$, which makes the nodes less distinguishable. Meanwhile, the embeddings generated from our Diverse Message Passing will converge to $(1,0,0)$, $(1,1,0)$, $(1,1,1)$, $(0,1,0)$, $(0,1,1)$, $(0,0,1)$ and $(1,0,1)$, which gives more diverse results. When $F$ increases, the advantage of our Diverse Message Passing becomes more significant.

## 4 Evaluations

In this section, the proposed Diverse Message Passing (DMP) is evaluated on real data from two perspectives: 1) handling networks with heterophily; 2) alleviating the over-smoothing issue.

### 4.1 Datasets

Our DMP is validated on 9 networks, which are shown in Table 1. These 9 networks can be categorized into 4 types of datasets. **Citation networks**: Cora, Citeseer, and Pubmed, which are widely used to evaluate GNNs, are the standard citation network benchmark datasets [33, 34]. In these networks, nodes and edges represent papers and citations among them, respectively. Words in the paper are employed to represent the node features in the bag-of-word form. The academic topic of each paper is taken as the label of node. **WebKB webpage networks**: Cornell, Texas, and Wisconsin are the webpage networks which are captured from the computer science departments of these universities, respectively. In these network, nodes and edges respectively represent the webpages and hyperlinks among them. Similar to the Citations networks, words in the webpage are employed to represent the node features in the bag-of-word form. The webpages are manually classified into five categories, student, project, course, staff and faculty. **Co-occurrence network**: Actor network contains the co-occurrences of actors in films, which are extracted from the heterogeneous information networks. It describes the complex relationships among films, directors, actors and writers [35]. In this network, nodes and edges stand for actors and their co-occurrences in films, respectively. The actor's Wikipedia page is exploited to extract features and node labels. **Wikipedia networks**:

Table 1: Datasets

| Dataset | Cora | Citeseer | Pubmed | Cham. | Squirrel | Actor | Cornell | Texas | Wisconsin |
|---|---|---|---|---|---|---|---|---|---|
| # Nodes | 2,708 | 3,327 | 19,717 | 2,277 | 5,201 | 7,600 | 183 | 183 | 251 |
| # Edges | 5,429 | 4,732 | 44,338 | 36,101 | 217,073 | 33,544 | 295 | 309 | 499 |
| # Features | 1,433 | 3,703 | 500 | 2,325 | 2,089 | 931 | 1,703 | 1,703 | 1,703 |
| # Classes | 7 | 6 | 3 | 5 | 5 | 5 | 5 | 5 | 5 |
| Homphily | 0.83 | 0.71 | 0.79 | 0.25 | 0.22 | 0.24 | 0.11 | 0.06 | 0.16 |

Chameleon and Squirrel are the webpages extracted from different topics in Wikipedia [36]. Similar to WebKB, nodes and edges respectively denote the webpages and hyperlinks among them, and informative nouns in the webpages are employed to construct the node features in the bag-of-word form. Webpages are classified in term of the averaged number of the monthly traffic. In Table 1, the network homophily rates, as computed in [20], are provided. It shows that the citation networks often possess high homphily rate, while others only possess low homophily rate, i.e., they are the networks with heterophily.

Besides, three heterogenous information networks (HINs), i.e., DBLP, ACM and IMDB, are also employed [37]. Different from the homogeneous networks shown in Table 1, HINs possess multiple types of nodes or edges, and multiple meta-paths can be utilized. **DBLP** contains 14,328 papers, 4,057 authors, 20 conferences and 8,789 terms. The authors are classified into four fields. **ACM** consists of 3,025 papers, 5,835 authors and 56 subjects. Papers are categorized into three fields (Database, Wireless Communication and Data Mining). **IMDB** contains 4,780 movies, 5,841 actors and 2,269 directors. The movies are classified into three classes (Action, Comedy and Drama) according to their genres. Note that the node features are in the bag-of-word form by extracting words from papers/plots.

### 4.2 Baselines

To demonstrate the superiority of DMP, 12 state-of-the-art methods are adopted as the baselines. Three of them, i.e., GCN [3], GAT [18] and GraphSAGE [22] are the standard GNNs, which achieve decent performances on networks with high homophily. ChebNet [24] is a classic spectral GNN. GCN, ChebNet and GraphSAGE with JKNet, which combines the multi-scale information via residual connections. MixHop [11] combines multi-scale information by concatenation. H2GCN [8] shows that concatenation is the key to process the networks with heterophily. Geom-GCN [20] attempts to leverage network embeddings in GNNs for processing the networks with homophily/heterophily.

### 4.3 Implementation Details

For all the datasets, nodes in each class are randomly split into three groups, 48% for training, 32% for validation, and 20% for testing, as mentioned in [8]. The performances of all the models are obtained by computing the averaged results over 10 random splits. The hyper-parameters, including weight decay, dropout, initial learning rate and patience for learning rate decay, are tuned by searching on the validation set. Adam [38] is adopted as the optimizer for all the models. For fair comparisons to GCN and GAT, standard DMP utilizes a two-layered model.

There exists many different implementations of our DMP. The two strategies introduced in Section 3.2 are named as DMP-1 (Eqs. (4) and (5)) and DMP-2 (Eqs. (6) and (7)), respectively. DMP-Deg stands for the degenerated DMP introduced in Eq. (8) in Section 3.3. Besides, two implementations, which consider the combining approach of first-order and second-order topology information, is also realized, i.e., Concatenation in H2GCN [8] and GraphSage [22] (denoted as "-Con"), and Summation in GCN [3] and GAT [18] (denoted as "-Sum"). To demonstrate the effectiveness of negative propagation weights in the above DMP implementations, positive weight learning, which replaces $tanh(\cdot)$ in Eq. (5) with $softmax(\cdot)$, is denoted as "-Posi".

### 4.4 Supervised Node Classification

The results of supervised node classification are shown in Table 2. In general, DMP achieves comparable performances on networks with high homophily, such as Cora, CiteSeer and Pubmed,

Table 2: Mean Classification Accuracy (Bold indicates the best, italics indicates the second best).

| Methods | Texas | Wisconsin | Actor | Squirrel | Cham. | Cornell | Citeseer | Pubmed | Cora |
|---------|-------|-----------|-------|----------|-------|---------|----------|--------|------|
| GraphSAGE | 82.43 | 81.18 | 34.23 | 41.61 | 58.73 | 75.95 | 76.04 | 88.45 | 86.90 |
| GCN | 64.86 | 56.86 | 31.12 | 32.28 | 53.51 | 54.05 | 75.53 | 84.71 | 85.51 |
| GAT | 58.38 | 55.29 | 26.28 | 30.62 | 54.69 | 58.92 | 75.46 | 84.68 | 82.68 |
| SAGE+JK | 83.78 | 81.96 | 34.28 | 40.85 | 58.11 | 75.68 | 76.05 | 88.34 | 85.96 |
| ChebNet+JK | 78.38 | 82.55 | 35.14 | *45.03* | **63.79** | 74.59 | 74.98 | 89.07 | 85.49 |
| GCN+JK | 66.49 | 74.31 | 34.18 | 40.45 | *63.42* | 64.59 | 74.51 | 88.41 | 85.79 |
| GCN-ChebNet | 77.30 | 79.41 | 34.11 | 43.86 | 55.24 | 74.32 | 75.82 | 88.72 | 86.76 |
| MixHop | 77.84 | 75.88 | 32.22 | 43.80 | 60.50 | 73.51 | 76.26 | 85.31 | *87.61* |
| GEOM-GCN | 67.57 | 64.12 | 31.63 | 38.14 | 60.90 | 60.81 | **77.99** | **90.05** | 85.27 |
| H2GCN | 84.86 | *86.67* | **35.86** | 36.42 | 57.11 | 82.16 | *77.04* | *89.40* | 86.92 |
| GPRGNN | **90.49** | 85.33 | 34.17 | 32.44 | 51.63 | **91.14** | 74.07 | 88.27 | **88.14** |
| DMP-Deg | 78.38 | 80.39 | 33.09 | 32.46 | 54.38 | 83.78 | 76.87 | 88.10 | 86.31 |
| DMP-2-Sum | 78.37 | 84.31 | 34.93 | 32.18 | 55.92 | *83.78* | 76.27 | 88.15 | 85.31 |
| DMP-2-Con | 83.78 | 84.31 | 34.67 | 44.28 | 60.53 | *83.78* | 75.97 | 85.31 | 85.31 |
| DMP-1-Posi | *86.48* | 84.31 | *35.72* | 34.96 | 51.53 | 70.27 | 75.67 | 88.10 | 86.11 |
| DMP-1-Sum | 86.48 | 86.27 | 34.21 | 43.42 | 50.21 | 70.27 | 76.13 | 88.13 | 82.28 |
| DMP-1-Con | *89.19* | **92.16** | 35.06 | **47.26** | 62.28 | *89.19* | 76.43 | 89.27 | 86.52 |

compared to all the state-of-the-art baselines. Meanwhile DMP significantly outperforms the baselines, especially the methods designed for processing the network with heterophily (such as H2GCN and Geom-GCN), on networks with heterophily (such as Texas, Wisconsin and Squirrel). Specifically, compared to H2GCN, which is the SOTA for processing the networks with heterophily, the obvious improvements on Texas, Wisconsin and Squirrel are 5.10%, 6.33% and 29.76%, respectively. This demonstrates the superiority of DMP in handling the networks with heterophily.

**Representation Ability:** 1) As elaborated in Section 3.2, DMP-2 is the simplification of DMP-1, and DMP-Deg is the degenerated version of DMP-1. Then, the ideal representation abilities are monotonically decreasing for DMP-1, DMP-2 and DMP-Deg. The results in Table 2 have verified the representation abilities of these methods. The performances of DMP-1 are higher than that DMP-2, and both of them give much higher performances than DMP-Deg. The performance gaps are more remarkable on networks with heterophily, since these networks require GNNs to possess powerful representation ability to exploit the informative attributes. 2) GraphSAGE, MixHop and H2GCN have shown that the concatenation strategy possesses high representation ability than the summation strategy employed in GCN and GAT. This strategy is also employed in the implementation of DMP. In Table 2, the accuracies of "-Con"'s are higher than those of "-Sum"'s. 3) The superiority of allowing negative propagation weights is also validated by the improvements of DMP-1-Con over DMP-1-Posi, especially on the networks with heterophily. Therefore, DMP-1-Con is exploited as the final DMP implementation, which will be used in the following experiments.

### 4.5 Semi-supervised Classification on Heterogenous Information Networks

Since only one type of nodes are labelled in HINs, the effectiveness of DMP is verified on the nodes with given labels. For the semi-supervised node classification task, 20% of the nodes in each class are utilized for training. To demonstrate the universality, DMP ignores the node types and meta-paths used in HAN [37] and treats the networks as homogenous networks. Two implementations of DMP, i.e., DMP-1-Con and DMP-1-Sum, are compared to GAT [18] and HAN [37]. The classifica-

Table 3: Accuracy on HINs in terms of Macro-F1 Score.

| Methods | DBLP | ACM | IMDB |
|---------|------|-----|------|
| GAT | 90.97 | 86.23 | 49.44 |
| HAN | **92.83** | 90.96 | 56.77 |
| DPM-1-Sum | 84.56 | 92.62 | **58.58** |
| DPM-1-Con | 92.49 | **92.64** | 57.73 |

tion accuracies in terms of Macro-F1 Score are reported in Table 3. As can be observed, HAN outperforms GAT by manually selecting the semantic meta-paths and combining the results from multiple meta-paths. Although the proposed DMP ignores the semantic meta-path, it can still achieve comparable or better performances compared to HAN. The main reason of this surprising results is that DMP can explore more semantic information and learns the meta-paths instead of manually

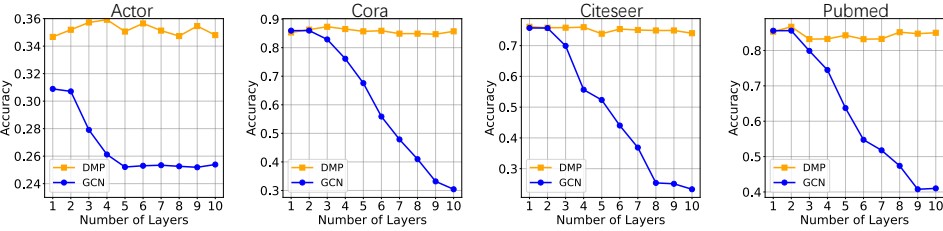

Figure 2: Classification accuracy results with various model depths.

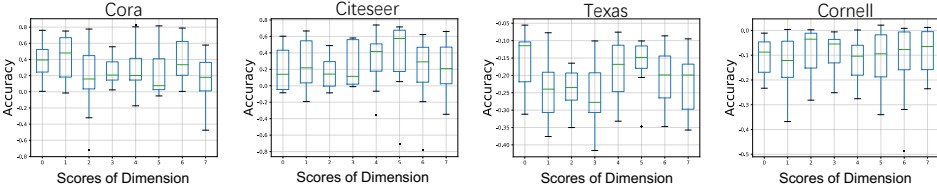

Figure 3: Distributions of the learned weights of sampled attribute dimensions.

selecting. Specifically, since HINs consist of multiple nodes or edges, the mutual impacts between different kinds of nodes tend to be different. These differences can not only be manually utilized by selection semantic meta-paths as HAN, but also be automatically exploited in DMP, which allows diverse attribute propagations. Apparently, the results demonstrate the superior representation ability of DMP.

### 4.6 Alleviating the Over-smoothing Issue

In Section 3.4, the potential on alleviating the over-smoothing issue is theoretically analyzed. In this section, this potential is experimentally verified on networks with homophily. The performance comparisons between DMP and GCN with various model depths are shown in Figure 2. As can be observed, when the number of layers increases, the performances of GCN tend to significantly degrade, while the performances of DMP remain stable. This trend verifies that DMP can effectively alleviate the over-smoothing issue.

### 4.7 Visualization of Learned Weights

To provide an intuitive understanding, this section provides the visualization of the distributions of learned weights of each attribute dimension. Due to all networks possess high attribute dimension, only 8 attribute dimensions are sampled for each network. For each attribute dimension, different edges obtain different propagation weights via learning scheme. Figure 3 visualizes the distributions of learned weights of sampled attribute dimensions. Most learned weights in networks with homophily, such as Cora and CiteSeer, are positive, while those in networks with heterophily, such as Texas and Squirrel, are positive. This also meets the definitions of networks with homophily and heterophily.

## 5  Conclusions

This paper investigates the attribute homophily rate and its impacts on the design of GNNs. Most of the existing GNNs perform *uniform* message passing by ignoring this factor. However, statistics on networks have revealed that attributes usually possess diverse homophily. To better exploit this observation, a Diverse Message Passing (DMP) framework, which specifies each attribute propagation weight along every edge, is proposed, and two specific strategies are proposed to reduce computational complexity. A theoretical analysis is provided to show that Diverse Message Passing may break the ceiling of representation ability of the spectral GNNs, as well as alleviate the common over-smoothing issue. Evaluations on real networks have verified the superiority of Diverse Message Passing on handling the networks with heterophily and alleviating the over-smoothing issue, compared to the state-of-the-art methods.

## Acknowledgments and Disclosure of Funding

This work was supported in part by the National Natural Science Foundation of China under Grant 61972442, Grant 61802391, Grant 62102413, Grant U2001202, Grant U1936208 and Grant 61802282, in part by the Key Research and Development Project of Hebei Province of China under Grant 20350802D and 20310802D, in part by the Natural Science Foundation of Hebei Province of China under Grant F2020202040, in part by the Natural Science Foundation of Tianjin of China under Grant 20JCYBJC00650, in part by the Science and Technology Specialists Project from Science and Technology Program of Tianjin under Grant 20YDTPJC00670, and in part by State Key Laboratory of Software Development Environment under Grant SKLSDE-2020ZX-18.

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
