# OpenReview forum: "Diverse Message Passing for Attribute with Heterophily"
_NeurIPS.cc/2021/Conference — NeurIPS 2021 Poster_

### Official Review · Reviewer_brCz · 2021-07-16

**Rating:** 7
**Confidence:** 4

**Summary:**

This paper extends the node-level network homophily based on the node label to attribute-level one by considering each attribute as a weak label. Then, it observes that different attributes have diverse attribute homophily rates. To alleviate the issue of existing uniform message passing on capturing this diverse attribute homophily rates, this paper proposes diverse message passing, where different attributes are assigned different propagation weights, and two implementations. Finally, it theoretically investigates the ability of diverse message passing on overcoming over-smoothing by analyzing the roles of the uniform and diverse propagation weights.

**Limitations And Societal Impact:**

Yes

**Main Review:**

The extension from network homophily rate to attribute homophily rate makes sense since node attributes can be seen as the weak label to represent the node compared to the node label. It is interesting to separately consider the homophily rate of each attribute and propose diverse propagation weights for different attributes. By comparing the proposed two implementation strategies and the interpretation from the spectral perspective, the two strategies can be regarded as a compromise between the expressive power and the overfitting issue. Thus, I think the proposed diverse message passing is effective in capturing the diverse homophily rate of different attributes. Therefore, this paper is technically sound.

Besides, the theoretical contribution of this paper is significant. Different from previous works, which often consider the over smoothing issue is caused by the over propagation, this paper shows that the over smoothing issue is actually induced by the uniform propagation. The theorems show an interesting finding that the propagation weight learning is equivalent to graph partition, and connect the graph partition and over smoothing issue. Thus, the theoretical analysis provides a new perspective to understand the over smoothing issue.

The experimental evaluation is sufficient. It demonstrates the superior performance on networks with heterophily and heterogeneous information networks and the ability on preventing the over-smoothing issues.

Although I think this paper is interesting and novel, there are also some concerns.
1.	Why the performance improvements on networks with heterophily are more significant than those on networks with homophily?
2.	The explanation of why multiple graph partition just below theorem 5 is hard to understand. It is better to reorganize this paragraph.
3.	The description of the data in Figure 1 is not very clear in the appendix.
The performance of the proposed DMP can’t outperform some SOTA methods designed for heterogeneous such as MAGNN.


**Time Spent Reviewing:**

4

---

> ### Author Response · Authors · 2021-08-10
> **Clarifying the performance improvement and Theorem  1**
>
> We would like to express our sincere appreciations to the reviewers for their insightful comments and compliments to our paper.
>
>
>
> ### Q1. Why the performance improvements on networks with heterophily are more significant than those on networks with homophily?
>
> R1. This mainly attributes to the different diversity levels of homophily rates of attributes on different networks. The insignificant performance improvements on datasets with high homophily mainly attribute to that “most attributes possess high homophily rates on networks with high homophily as shown in Figure 2, and this can be fit by existing GNNs, which actually perform smoothing-based uniform message passing. However, since the diversities of homophily rates of attributes are more remarkable on networks with low homophily, the proposed diverse message passing can achieve significant performance improvements on them. Therefore, the proposed method often achieves remarkable performance on networks with low homophily rates.
>
>
> ### Q2. Why multiple graph partition can alleviate over-smoothing issue.
>
> According to [R1], the over-smoothing issue is the cause that all the node embeddings tend to converge to a few vectors, which makes the nodes indistinguishable.
>
>
> In the following example, we will show the number of converged vectors from our Diverse Message Passing is much larger than that from Uniform Message Passing, which indicates that our Diverse Message Passing can effectively alleviate the over-smoothing issue.
> Based on the smoothing characteristics of graph convolutional layers, Theorem 1 in [R1] proves that, as the number of graph convolutional layers increases, the node embeddings will converge to a few vectors, which is the over-smoothing issue. These vectors are fully determined by the indication vectors of the connected components. Specifically, given a graph $\mathcal{G}$ with $k$ connected components $\{C_i\}_{i=1}^{k}$, the indication vector for the $i$-th component is denoted by $\mathbf{1}^{(i)}\in\\{0,1\\}^k$. This vector indicates whether a vertex is in the component $C_i$, i.e.,$\mathbf{1}^{(i)}_j=1$ if  $v_j \in C_i$; $\mathbf{1}^{(i)}_j=0$ if $v_j \not\in C_i$.
>
>
> According to Theorem 4 in our paper, the Uniform Message Passing actually partitions the graph $\mathcal{G}$ into F connected components. Then, the embeddings in Uniform Message Passing will converge to $\mathbf{1}^{(i)}\in\\{0,1\\}^k$, where only one element is 1 and other element are 0, for $i=1,…,F$, i.e., the over-smoothing issue appears.
>
>
> Theorem 5 indicates that our Diverse Message Passing generates F groups of 2-components partitions, which is equivalent to generating $2^F$ candidate components. If the 2-components partition of the j-th group for node $i$ is represented by a scalar $t_j^{(i)} = \\{0,1\\}$, the F groups of 2-components partitions for node $i$ can be represented by $\mathbf{t}^{(i)} = \\{0,1\\}^k$. Note that different from $\mathbf{1}^{(i)}\in\\{0,1\\}^k$ in Uniform Message Passing, where only one element is 1 and the other elements are 0, each element in $\mathbf{t}^{(i)} = \\{0,1\\}^k$ can be either 1 or 0. Then, $\mathbf{t}^{(i)} = \\{0,1\\}^k$ may possess $2^F$ different values. Therefore, the embeddings in our Diverse Message Passing converge to $\mathbf{t}^{(i)} = \\{0,1\\}^k$.
>
>
> The number of different values in $\mathbf{t}^{(i)} = \\{0,1\\}^k$ is $2^F$, which is much larger than that in $\mathbf{1}^{(i)}\in\\{0,1\\}^k$, i.e., $F$. Thus, the number of converged values from our Diverse Message Passing is much larger than that from Uniform Message Passing, which proves that our Diverse Message Passing can effectively alleviate the over-smoothing issue.
>
>
> For intuitive illustration, a toy example with $F=3$ is provided here.
> The embeddings generated from Uniform Message Passing will converge to (1,0,0), (0,1,0) and (0,0,1), which makes the nodes indistinguishable. In contrast, the embeddings generated from our Diverse Message Passing will converge to (1,0,0), (1,1,0), (1,1,1), (0,1,0), (0,1,1), (0,0,1) and (1,0,1), which gives more diverse results. As F increase, the advantage of our Diverse Message Passing becomes more significant.
>
>
>
>
> ### Q3. The performance of the proposed DMP can’t outperform some SOTA method designed for heterogeneous such as MAGNN.
>
> R3. This may be caused by that most SOTA methods designed for HINs fully explore the semantic meta-paths information, while our proposed DMP only employs the original topology information. It is worth noting that the proposed DMP has outperformed HAN, which also adopts meta-path information. In the future, we will consider how to incorporate meta-path into the DMP for better performance on HINs.
>
>
>
> [R1] Qimai Li, Zhichao Han, Xiao-Ming Wu: Deeper Insights Into Graph Convolutional Networks for Semi-Supervised Learning. AAAI 2018: 3538-3545

---

### Official Review · Reviewer_RdEt · 2021-07-16

**Rating:** 4
**Confidence:** 4

**Summary:**

The authors extend network homophily rate to attribute homophily rate by taking attribute as weak label and design diverse message passing (DMP) framework to specifiy each attribute propagation weight along each edge. The authors show that DMP can prevent  over-smoothing and handle network with heterophily.


**Limitations And Societal Impact:**

Yes, the authors adequately addressed the limitations and potential negative societal impact of their work.

**Main Review:**

Strength:

1. Although DMP looks like a simple extension of GAT, I think to form uniform and diverse message passing into an optimization problem is interesting.
2. The visualization looks good.

Weakness&Advice:

1. Line 74, “ddimensions” —> “dimensions”

2. In equation (3), is it possible to get a zero denominator? Do you assume all elements $x_{vf}$ are positive?

3. In equation (4)(6), DMP looks like a channel-wise GAT with attention values in (-1,1). The contribution and novelty in section 3.2 are limited.

4. In theorem 5, should add $f=1.\dots, F$

5. Line 213-216, “different form Uniform Message Passing, which directly generates a F-components partition, Diverse Message Passing generates F groups 2-components partitions as candidates and then the  classifier in semi-supervised task determines how to combine them to form the final F-components partition.” For shallow GNNs, why generating  F groups of 2-components partitions is better than generating one F-components partition on semi-supervised learning. Need more explanation.

6. The dimension of $X$ is $N\times F$ and F is number of partitions you use in section 3.4. What is the relation between the two $F$ or do I miss something? $F$ is an assumption or a restriction for theorem 4?

7. Making correction of theorem 5 in supplementary material seems not to be allowed.

8. Should add a comparison with GPRGNN, which is a new model to solve heterophily problem.

9. Standard deviation should be provided in table 2&3.

10. There are two strategies proposed to reduce the number of learnable parameters. So what is the running speed of the current models and how they compare with the existing methods?

11. How well does DMP on solving over-smoothing compared to the existing models.


**Time Spent Reviewing:**

8

---

> ### Author Response · Authors · 2021-08-10
> **Clarifying the technical contribution of this work and the Theorem 5**
>
> ### Q1. In equation (4)(6), DMP looks like a channel-wise GAT with attention values in (-1,1). The contribution and novelty in section 3.2 are limited.
>
> R1. Although the attention mechanism is adopted, we want to emphasize that it is NOT our main technical contribution.
> 1. Our main technical contribution is the diverse message passing framework to model the diverse homophily rates of attributes.
> 2. The channel-wise GAT in the first strategy (Eq. (5)) is only an implementation of diverse message passing, and it can be replaced with other methods to compute the diverse propagation weights, such as [1].
> 3. The second strategy in Eqs. (6) and (7) is very different from the existing attention mechanism in graph learning. It computes propagation weight vector for each node by leveraging attention between node embedding and average embeddings of its neighborhoods.
>
> Therefore, we think the technical contribution of the proposed diverse message passing is remarkable.
> [1] Kai Zhang, Yaokang Zhu, Jun Wang, Jie Zhang: Adaptive Structural Fingerprints for Graph Attention Networks. ICLR 2020
>
>
> ### Q2. For shallow GNNs, Why generating F groups of 2-components partitions is better than generating one F-components partition on semi-supervised learning.
> R2.
> Thank you for your comments. We may not express ourselves clearly. In fact, the analysis is given for DEEP GNNs instead of the shallow ones. For the shallow ones, the advantage of our Diverse Message Passing comes from its capability in modeling the diverse homophily rates of the attributes. Here, we give more elaborations on the advantage of generating F groups of 2-components partitions for the semi-supervised learning task in DEEP GNNs.
>
> For (semi-)supervised learning task, to make the learned model robust, the embeddings of data should be diverse to represent the structure of each class. For example, if $x_1^c, x_2^c, …, x_{n_c}^c$ are the training samples from class $c$, $x_j^c$’s should be different to others, to better represent the structure of class $c$. For the extreme case where all the $x_j^c$’s are identical, training with t samples is as bad as that with one sample. Since the embeddings generated from F groups of 2-components partitions are more diverse than that from a single F-components partition in the semi-supervised learning task, we design our DMP by forming F groups of 2-components partitions.
> The diversities of the above two cases are analyzed as follows.
>
>
>
>
> Theorem 1 in [R1] proves that, as the number of graph convolutional layers increases, the node embeddings will converge to a few vectors, which is the over-smoothing issue. These vectors are fully determined by the indication vectors of the connected components. Specifically, given a graph $\mathcal{G}$ with $k$ connected components $\\{C_i\\}_{i=1}^{k}$, the indication vector for the $i$-th component is denoted by $\mathbf{1}^{(i)}\in\\{0,1\\}^k$. This vector indicates whether a vertex is in the component $C_i$, i.e., i.e.,$\mathbf{1}^{(i)}_j=1$ if  $v_j \in C_i$; $\mathbf{1}^{(i)}_j=0$ if $v_j \not\in C_i$.
>
>
> According to Theorem 4 in our paper, the Uniform Message Passing actually partitions the graph $\mathcal{G}$ into F connected components. Then, the embeddings in Uniform Message Passing will converge to $\mathbf{1}^{(i)}\in\\{0,1\\}^k$, where only one element is 1 and other elements are 0, for $i=1,…,F$, i.e., the over-smoothing issue appears.
>
> Theorem 5 indicates that our Diverse Message Passing generates F groups of 2-components partitions, which is equivalent to generating $2^F$ candidate components. If the 2-components partition of the j-th group for node $i$ is represented by a scalar $t_j^{(i)} = \\{0,1\\}$, the F groups of 2-components partitions for node $I$ can be represented by $\mathbf{t}^{(i)} = \\{0,1\\}^k$. Note that different from $\mathbf{1}^{(i)}\in\\{0,1\\}^k$ in Uniform Message Passing, where only one element is 1 and the other elements are 0, each element in $\mathbf{t}^{(i)} = \\{0,1\\}^k$ can be either 1 or 0. Then, $\mathbf{t}^{(i)} = \\{0,1\\}^k$ may possess $2^F$ different values. Therefore, the embeddings in our Diverse Message Passing converge to $\mathbf{t}^{(i)} = \\{0,1\\}^k$.
>
>
> In summary, the numbers of converged vectors in F groups of 2-components partitions and one F-components partition are $2^F$ and $F$, respectively.
> Thus, the embeddings obtained from F groups of 2-components partitions are more diverse than those from one F-components partition. Therefore, generating F groups of 2-components partitions is better than generating one F-components partition on semi-supervised learning.
>
>
> [R1] Qimai Li, Zhichao Han, Xiao-Ming Wu: Deeper Insights Into Graph Convolutional Networks for Semi-Supervised Learning. AAAI 2018: 3538-3545
>
>
> ### Q3. Comparison with GPRGNN.
> R3. The Mean Classification Accuracies of both proposed DMP and GPRGNN are given as follows.
>
> Method&nbsp;&nbsp;&nbsp; cora&nbsp;&nbsp; &nbsp;citeseer &nbsp;&nbsp; pubmed&nbsp;&nbsp; texas &nbsp;&nbsp;&nbsp; cornell &nbsp;&nbsp;&nbsp; film &nbsp;&nbsp;&nbsp;&nbsp;&nbsp; chameleon &nbsp;&nbsp;&nbsp; squirrel
>
> GPRGNN&nbsp; 88.14&nbsp;&nbsp;&nbsp; 74.07&nbsp;&nbsp;&nbsp;&nbsp;&nbsp;&nbsp;&nbsp; 88.27&nbsp;&nbsp;&nbsp;&nbsp;&nbsp;&nbsp; 90.49&nbsp;&nbsp;&nbsp;&nbsp;&nbsp;&nbsp; 91.14&nbsp;&nbsp;&nbsp; &nbsp;&nbsp;&nbsp;34.17&nbsp;&nbsp;&nbsp;&nbsp;&nbsp;&nbsp; &nbsp;&nbsp;51.63&nbsp;&nbsp;&nbsp;&nbsp;&nbsp;&nbsp;&nbsp;&nbsp;&nbsp;&nbsp;&nbsp; 32.44
>
> DMP&nbsp;&nbsp;&nbsp;&nbsp;&nbsp;&nbsp;&nbsp;&nbsp; 86.52&nbsp;&nbsp; &nbsp;76.43&nbsp;&nbsp;&nbsp;&nbsp;&nbsp;&nbsp;&nbsp; 89.27&nbsp;&nbsp;&nbsp;&nbsp;&nbsp;&nbsp; 89.19&nbsp;&nbsp;&nbsp; &nbsp;&nbsp;&nbsp;89.19&nbsp;&nbsp;&nbsp;&nbsp;&nbsp;&nbsp; 35.06&nbsp;&nbsp;&nbsp;&nbsp;&nbsp;&nbsp; &nbsp;&nbsp;62.28&nbsp;&nbsp;&nbsp; &nbsp;&nbsp;&nbsp;&nbsp;&nbsp;&nbsp;&nbsp;&nbsp;47.26
>
> Thus, the proposed DMP outperform GPRGNN on five datasets among all eights dataset. Especially, the improvements on networks with low homophily are more significant.
>
> ### Q4. What is the running speed of the current models and how they compare with the existing methods?
>
> R4. According to your suggestion, we compare the speed of the proposed DMP with some other methods designed for networks with heterophily as follows, especially the SOTA method H2GCN. The results are the runtime to train and test the model in terms of seconds.
>
> Dataset&nbsp;&nbsp;&nbsp;&nbsp; &nbsp;&nbsp;&nbsp; DMP&nbsp;&nbsp;&nbsp;&nbsp;&nbsp; JKNet&nbsp;&nbsp;&nbsp; ChebNet &nbsp;&nbsp;&nbsp;H2GCN
>
> cornell&nbsp;&nbsp;&nbsp;&nbsp;&nbsp;&nbsp;&nbsp; &nbsp;&nbsp;&nbsp;18.87&nbsp;&nbsp;&nbsp;&nbsp;&nbsp;&nbsp; 8.44 &nbsp;&nbsp;&nbsp;10.56&nbsp;&nbsp;&nbsp;&nbsp;&nbsp;&nbsp; 128.45
>
> texas&nbsp;&nbsp;&nbsp;&nbsp;&nbsp;&nbsp;&nbsp;&nbsp;&nbsp;&nbsp;&nbsp;&nbsp;&nbsp; 18.53&nbsp;&nbsp;&nbsp; 16.43&nbsp;&nbsp;&nbsp;&nbsp;&nbsp;&nbsp; 11.41&nbsp;&nbsp;&nbsp;&nbsp;&nbsp;&nbsp; 83.06
>
> film&nbsp;&nbsp;&nbsp;&nbsp;&nbsp;&nbsp;&nbsp;&nbsp; &nbsp;&nbsp;&nbsp;&nbsp;&nbsp;&nbsp; 19.13 &nbsp;&nbsp;&nbsp;&nbsp;9.45 &nbsp;&nbsp;&nbsp;&nbsp;&nbsp;&nbsp;16.34 &nbsp;&nbsp;&nbsp;&nbsp;&nbsp;&nbsp;363.03
>
> chameleon &nbsp;20.60 &nbsp;&nbsp;&nbsp;&nbsp;&nbsp;25.16&nbsp;&nbsp;&nbsp;&nbsp;&nbsp;&nbsp; 31.90&nbsp;&nbsp;&nbsp;&nbsp;&nbsp; 226.78&nbsp;&nbsp;&nbsp;&nbsp;&nbsp;&nbsp;
>
> citeseer &nbsp;&nbsp;&nbsp;&nbsp;&nbsp;&nbsp;&nbsp;20.70&nbsp;&nbsp;&nbsp;&nbsp;&nbsp; 20.74&nbsp;&nbsp;&nbsp;&nbsp;&nbsp;&nbsp; 28.95&nbsp;&nbsp;&nbsp;&nbsp; 116.09
>
> cora&nbsp;&nbsp;&nbsp;&nbsp; &nbsp;&nbsp;&nbsp;&nbsp;&nbsp;&nbsp;21.79 &nbsp;&nbsp;&nbsp;&nbsp;&nbsp;&nbsp;22.29 &nbsp;&nbsp;&nbsp;&nbsp;&nbsp;&nbsp;27.72 &nbsp;&nbsp;&nbsp;&nbsp;&nbsp;&nbsp;288.23
>
> pubmed&nbsp;&nbsp;&nbsp; 18.24&nbsp;&nbsp;&nbsp;&nbsp;&nbsp;&nbsp; 20.99&nbsp;&nbsp;&nbsp;&nbsp;&nbsp;&nbsp; 42.66&nbsp;&nbsp;&nbsp;&nbsp;&nbsp;&nbsp; 340.25
>
> squirrel&nbsp;&nbsp;&nbsp;&nbsp; 30.39&nbsp;&nbsp;&nbsp;&nbsp;&nbsp;&nbsp; 27.76 &nbsp;&nbsp;&nbsp;&nbsp;&nbsp;&nbsp;60.54&nbsp;&nbsp;&nbsp;&nbsp;&nbsp;&nbsp; 869.14
>
>
> It can be observed that the proposed DMP is faster than others on large networks. Note that, DMP is faster than H2GCN on all the datasets.
>
>
> ### Q5. How well does DMP on solving over-smoothing compared to the existing models.
> R5. Thank you for your suggestion, and we will add the Classification accuracy results with various depths of other methods into Figure 2. The reasons why we omit them in the current version are as follows. First, it may be difficult to directly compare their ability to solve over-smoothing issue, since the performances of our proposed DMP and most existing methods to solve over-smoothing issue are stable as the number of layers increases. Second, the proposed DMP is mainly designed for networks with heterophily, and its potential to solve over-smoothing issue is just a byproduct.
>
>
> ### Q6. The dimension of $X \in R^{N \times F}$ and F is number of partitions. What is the relation between the two F.
> R6. Thank you for pointing out this issue. This confusion is caused by that F is employed to represent the dimensions of X and H in Sec. 2 and Sec 3.4, respectively. And we will correct this issue in the next version.
> In section 3.4, i.e., theory analysis, we set the dimension of H as $R^{N \times F}$ and provide the theorem that the number of partition is identical to the number of the dimension of H, i.e. F, in uniform message passing.
>
> ### Q7: Do you assume all elements x are positive?
> R7. Yes, we assume all elements x’s are non-negative, and this is true in all the datasets.
>
> ### Q8: Other issues and suggestion. Line 74, “ddimensions” —> “dimensions”. Standard deviation should be provided in table 2&3. In theorem 5, should add f=1…F.
> R8: We would like to express our sincere appreciation to you for the insightful comments. Except for the above responses, we will correct other issues raised by you.

---

### Official Review · Reviewer_x1nw · 2021-07-16

**Rating:** 6
**Confidence:** 4

**Summary:**

This paper proposes a diverse message passing framework (DMP) to generalize a classic uniform message passing framework and also suggests strategies preventing over-smoothing issues.

**Limitations And Societal Impact:**

The authors did not provide the “limitations and social impact” of this work in the paper.


**Main Review:**

=================================================================

[Main Strengths]

This paper's main strength is that the motivations of the work are well described (as shown in SEction 3.1) and evaluations across various homophily rate datasets (see Table1) are well designed.

=================================================================

[Main Weaknesses]

The paper's main weakness is the somewhat limited merits of the theoretical analysis shown in Section 3.4. In particular, the potential benefits of the contribution of Theorem 5 toward preventing an over-smoothing issue are not clear to me.

=================================================================

**Time Spent Reviewing:**

1.5 hour

---

> ### Author Response · Authors · 2021-08-10
> **Clarifying the potential benefits of Theorem 5 toward preventing an over-smoothing issue**
>
> We would like to express our sincere appreciations to the reviewers for their insightful comments and compliments to our paper.
>
>
> According to [R1], the over-smoothing issue is the cause that all the node embeddings tend to converge to a few vectors, which makes the nodes indistinguishable.
>
>
> In the following example, we will show the number of converged vectors from our Diverse Message Passing is much larger than that from Uniform Message Passing, which indicates that our Diverse Message Passing can effectively alleviate the over-smoothing issue.
> Based on the smoothing characteristics of graph convolutional layers, Theorem 1 in [R1] proves that, as the number of graph convolutional layers increases, the node embeddings will converge to a few vectors, which is the over-smoothing issue. These vectors are fully determined by the indication vectors of the connected components. Specifically, given a graph $\mathcal{G}$ with $k$ connected components $\{C_i\}_{i=1}^{k}$, the indication vector for the $i$-th component is denoted by $\mathbf{1}^{(i)}\in\\{0,1\\}^k$. This vector indicates whether a vertex is in the component $C_i$, i.e.,$\mathbf{1}^{(i)}_j=1$ if  $v_j \in C_i$; $\mathbf{1}^{(i)}_j=0$ if $v_j \not\in C_i$.
>
>
> According to Theorem 4 in our paper, the Uniform Message Passing actually partitions the graph $\mathcal{G}$ into F connected components. Then, the embeddings in Uniform Message Passing will converge to $\mathbf{1}^{(i)}\in\\{0,1\\}^k$, where only one element is 1 and other element are 0, for $i=1,…,F$, i.e., the over-smoothing issue appears.
>
>
> Theorem 5 indicates that our Diverse Message Passing generates F groups of 2-components partitions, which is equivalent to generating $2^F$ candidate components. If the 2-components partition of the j-th group for node $i$ is represented by a scalar $t_j^{(i)} = \\{0,1\\}$, the F groups of 2-components partitions for node $i$ can be represented by $\mathbf{t}^{(i)} = \\{0,1\\}^k$. Note that different from $\mathbf{1}^{(i)}\in\\{0,1\\}^k$ in Uniform Message Passing, where only one element is 1 and the other elements are 0, each element in $\mathbf{t}^{(i)} = \\{0,1\\}^k$ can be either 1 or 0. Then, $\mathbf{t}^{(i)} = \\{0,1\\}^k$ may possess $2^F$ different values. Therefore, the embeddings in our Diverse Message Passing converge to $\mathbf{t}^{(i)} = \\{0,1\\}^k$.
>
>
> The number of different values in $\mathbf{t}^{(i)} = \\{0,1\\}^k$ is $2^F$, which is much larger than that in $\mathbf{1}^{(i)}\in\\{0,1\\}^k$, i.e., $F$. Thus, the number of converged values from our Diverse Message Passing is much larger than that from Uniform Message Passing, which proves that our Diverse Message Passing can effectively alleviate the over-smoothing issue.
>
>
> For intuitive illustration, a toy example with $F=3$ is provided here.
> The embeddings generated from Uniform Message Passing will converge to (1,0,0), (0,1,0) and (0,0,1), which makes the nodes indistinguishable. In contrast, the embeddings generated from our Diverse Message Passing will converge to (1,0,0), (1,1,0), (1,1,1), (0,1,0), (0,1,1), (0,0,1) and (1,0,1), which gives more diverse results. As F increase, the advantage of our Diverse Message Passing becomes more significant.
>
>
> [R1] Qimai Li, Zhichao Han, Xiao-Ming Wu: Deeper Insights Into Graph Convolutional Networks for Semi-Supervised Learning. AAAI 2018: 3538-3545

---

### Official Review · Reviewer_wnNV · 2021-07-16

**Rating:** 7
**Confidence:** 4

**Summary:**

This paper focused on improving graph neural networks from homophily and over-smoothing issues by introducing a diverse message passing (DMP) framework. Experiments on 9 public datasets with a variety of homophily scores were provided to validate the effectiveness of the proposed method.

**Limitations And Societal Impact:**

Suffice.

**Main Review:**

Pros:
-	Following previous works [8], the motivation of this work is clearly presented.  How the proposed method connects to existing GNN methods has been elaborated in both spatial and spectral domains.
-	While the proposed implementation is simple and straightforward, some theoretical analyses (Theorem 4 and 5, though with some key typos) have been provided to demonstrate the benefits of DMP over the uniform message passing.
-	Extensive experimental results have been provided in terms of different homophily graphs and model components. Plus, experiments on heterogenous graphs and deep GNNs have also been conducted.

Cons:
-	The technical contribution of this work is limited since the proposed method mainly follows the framework provided in [8] and adopts a similar implementation way to compute “attentions” over attributes as in [18,22,23].
-	Some claims are over-emphasized as the proposed method is clearly built on top of several theoretical frameworks [6-8, 28]. To name a few, “existing spectral GNNs are just equivalent to a degeneration of diverse message passing”, “diverse message passing may break the ceiling of spectral GNNs”, etc.
-	The proposed method might be ad hoc to the datasets with low homophily rates.

**Time Spent Reviewing:**

3.5

---

> ### Author Response · Authors · 2021-08-10
> **Clarifying the technical contribution of this work**
>
> We would like to express our sincere appreciations to the reviewers for their insightful comments and compliments to our paper.
>
> ### Q1. The proposed method mainly follows the framework provided in H2GCN [8] and adopts a similar implementation way to compute “attentions” over attributes.
>
> R1. The proposed DMP is very different from the H2GCN proposed in [8], although [8] motivates us to consider the GNN for networks with heterophily.
> 1. Their motivations are different. DMP mainly takes into consideration the node attributes and finds that the homophily rates of different attributes vary significantly.  H2GCN mainly takes into consideration the graph topology, and finds that ego embedding, first-order neighbor embedding, and high-order neighborhood embedding should be separated.
> 2. Their implementations are different. To model the diverse homophily rates of attributes, DMP learns different propagation weights for different attribute channels. To separately model embeddings from different levels of the neighborhood, H2GCN concatenates ego embedding, first-order neighbor embedding, and high-order neighborhood embedding to form the final embedding.
>
> Besides, although “attention” over attribute is adopted, we want to emphasize that it is NOT our main technical contribution.
> 1. Our main technical contribution is the diverse message passing framework to model the diverse homophily rates of attributes.
> 2. The “attention” over attribute is only an implementation of diverse message passing, and it can be replaced with other methods, such as [1].
> 3. The second strategy in Eqs. (6) and (7) is very different from the existing attention mechanism in graph learning. It computes the propagation weight vector for each node by leveraging attention between node embedding and average embeddings of its neighborhoods.
>
>
> Therefore, we think the technical contribution of the proposed diverse message passing is remarkable.
> [1] Kai Zhang, Yaokang Zhu, Jun Wang, Jie Zhang: Adaptive Structural Fingerprints for Graph Attention Networks. ICLR 2020
>
>
> ### Q2. The proposed method might be ad hoc to the datasets with low homophily rates.
>
> R2. We think although the improvements of our proposed method are more significant on datasets with low homophily rates, it just corrects the biases of existing methods on networks with high homophily rates for the following reasons.
> 1.  As shown in Figure 1, the diversities of the homophily rates of attributes exist in networks with both high and low homophily. Thus, the proposed diverse message passing can be applied to all networks.  Figure 3 also shows that the learned propagation weights are diverse on all networks.
> 2. The insignificant performance improvements on datasets with high homophily mainly attribute to that “most attributes possess high homophily rates in networks with high homophily as shown in Figure 2, and this can be fit by existing GNNs, which actually perform smoothing-based uniform message passing.
> 3. Thus, to correct the uniform smoothing bias of existing GNNs, this paper proposes diverse message passing, which allows both smoothing (positive propagation weights) and sharpening (negative propagation weights). Since the diversities of homophily rates of attributes are more remarkable on networks with low homophily, the proposed diverse message passing can achieve significant performance improvements on them.
> Therefore, the proposed message passing is universal to all the datasets, although the performance improvements are more significant on datasets with low homophily rates.
>
>
> ### Q3. Some claims are over-emphasized.
>
> R3. Thank you very much. As you suggested, we will revise the draft as “the proposed diverse message passing may be more expressive than existing spectral GNNs.”

---

### Decision · Program_Chairs · 2021-09-27

**Decision:**

Accept (Poster)

**Comment:**

This paper proposes a message passing scheme for graph neural networks(GNNs) derived from homophily.  The problem is clearly motivated. The distinctness of the work is brought vis-a-vis GNN models has been elaborated in both spatial and spectral domains. One of the reviewers appreciated the theoretical insights which help in understand the shortcomings of existing literature. However the main concern seems to be that comparisons of the State of the art need to be improved.